# Observation of giant spin-split Fermi-arc with maximal Chern number in the chiral topological semimetal PtGa

Mengyu Yao [1,7], Kaustuv Manna [1,7 ✉], Qun Yang[1,7], Alexander Fedorov[2,3], Vladimien Voroshnin[2], B. Valentin Schwarze[4,5], Jacob Hornung[4,5], S. Chattopadhyay[4], Zhe Sun[6], Satya N. Guin [1], Jochen Wosnitza[4,5], Horst Borrmann[1], Chandra Shekhar [1], Nitesh Kumar [1], Jörg Fink[1,3,5], Yan Sun [1] & Claudia Felser [1 ✉]

Non-symmorphic chiral topological crystals host exotic multifold fermions, and their associated Fermi arcs helically wrap around and expand throughout the Brillouin zone between the high-symmetry center and surface-corner momenta. However, Fermi-arc splitting and realization of the theoretically proposed maximal Chern number rely heavily on the spin-orbit coupling (SOC) strength. In the present work, we investigate the topological states of a new chiral crystal, PtGa, which has the strongest SOC among all chiral crystals reported to date. With a comprehensive investigation using high-resolution angle-resolved photoemission spectroscopy, quantum-oscillation measurements, and state-of-the-art ab initio calculations, we report a giant SOC-induced splitting of both Fermi arcs and bulk states. Consequently, this study experimentally confirms the realization of a maximal Chern number equal to ±4 in multifold fermionic systems, thereby providing a platform to observe large-quantized photogalvanic currents in optical experiments.

[1] Max Planck Institute for Chemical Physics of Solids, 01187 Dresden, Germany. [2] Helmholtz-Zentrum Berlin fur Materialien und Energie, Berlin, Germany. [3] Institute for Solid State Research, Leibniz IFW Dresden, 01069 Dresden, Germany. [4] Dresden High Magnetic Field Laboratory (HLD-EMFL) and Würzburg-Dresden Cluster of Excellence ct.qmat, Helmholtz-Zentrum Dresden-Rossendorf, 01328 Dresden, Germany. [5] Institute for Solid-State and Materials Physics, Technical University Dresden, 01062 Dresden, Germany. [6] National Synchrotron Radiation Laboratory, University of Science and Technology of China, Hefei 230029, China. [7] These authors contributed equally: Mengyu Yao, Kaustuv Manna, Qun Yang. ✉email: Kaustuv.Manna@cpfs.mpg.de; Claudia.Felser@cpfs.mpg.de

The discovery of topological insulators reinvigorated the understanding of the electronic band structure and inspired generalization of the topological band theory concerning solid states[1–5]. This led to the discovery of quasi-particle excitations of the Dirac and Weyl fermions within solid-state materials characterized by a linear band crossing in metals along with the creation of a direct analogy between the said fermions and fundamental particles in high-energy physics[6–15]. On the other hand, quasiparticles within electronic band structures need not necessarily follow the Poincare symmetry pertaining to high-energy physics. Instead, they adhere to the crystal symmetry such that new types of fermionic excitations can be realized within solid states without having counterparts in high-energy physics[16–20].

Multifold fermions protected by chiral crystal symmetry attracted extensive attentions recently[16,21–26]. In comparison with Dirac fermion with zero topological charge and Weyl fermions with Chern number ±1, multifold fermions in chiral crystals host large Chern numbers and chiral Fermi arcs on their surface states (SSs). Since these symmetry-enforced multifold fermions locate at high-symmetry time-reversal invariant momenta, realization of long-surface Fermi arcs expanding throughout the Brillouin zone (BZ) becomes topologically guaranteed. These Fermi arcs are orders of magnitude larger and highly robust compared with those in any other Weyl semimetal. This affords a natural advantage over twofold degenerate Weyl fermions with regard to detection of Fermi-arc states. Identification of multifold fermions with large Chern numbers has previously been performed via observation of surface chiral Fermi arcs using angle-resolved photoemission spectroscopy (ARPES)[21–25] as well as the remarkable quantized circular photogalvanic effect[26].

According to theoretical predictions, the largest topological charge from multifold fermions has a Chern number ±4 hosted by compounds with space groups $P2_13$ (No. 198), especially in compounds such as CoSi, RhSi, PtAl, CoGe, RhGe, PdGa, etc.[16,21–24,26]. Protected by this topological Chern number of ±4, there exist four Fermi arcs crossing surface-projected multifold fermions at high-symmetry points. However, given the small spin splitting of Fermi arcs, all ARPES measurements performed to date have only confirmed the existence of chiral Fermi arcs connecting projected multifold fermions. That is, the exact number of Fermi arcs that exist has yet remained unclear. In other words, the theoretical Chern number of ±4 has so far not been experimentally verified by any surface-detection experiment. Realization of large spin splitting of the Fermi arc requires a very strong spin-orbit coupling (SOC) in the chiral crystals. Among all chiral multifold fermionic materials investigated thus far, PtGa demonstrates the strongest SOC. This paper reports results obtained using a combination of high-resolution ARPES, quantum oscillations, and state-of-the-art ab initio calculations to illustrate the giant spin splitting of topological states within a new chiral topological semimetal PtGa.

## Results

**Crystal growth and electrical transport measurement**. In this study, high-quality PtGa single-crystals were grown using the self-flux technique, as discussed in Methods and Supplementary Note 1. The crystal symmetry was confirmed via rigorous single-crystal diffraction analysis. The estimated Flack parameter value of 0.03 (4) indicates the existence of a single structural chirality in the crystal. The samples demonstrated metallic behavior throughout the measured temperature range (Supplementary Fig. 2). The observed large residual resistivity ratio (RRR = $\rho(300\,K)/\rho(2\,K)$) of ~84 and giant magnetoresistance of roughly 1000% at 2 K (Supplementary Fig. 2) reflect the high quality of the crystals. Values of the carrier concentration and mobility were estimated to be 2.1 × $10^{21}\,cm^{-3}$ and 4650 $cm^2\,V^{-1}\,s^{-1}$, respectively, using field-dependent Hall-resistivity data (Supplementary Fig. 3) and the longitudinal resistivity at 2 K.

**Crystal structure and electronic structure**. PtGa crystallizes in the non-symmorphic chiral space group $P2_13$ (no. 198) with lattice parameter $a = 4.9114(3)$ Å (Fig. 1a). The corresponding simple cubic BZ is depicted in Fig. 1c with high-symmetric momenta—Γ, X, M, and R. Results of ab initio calculations and

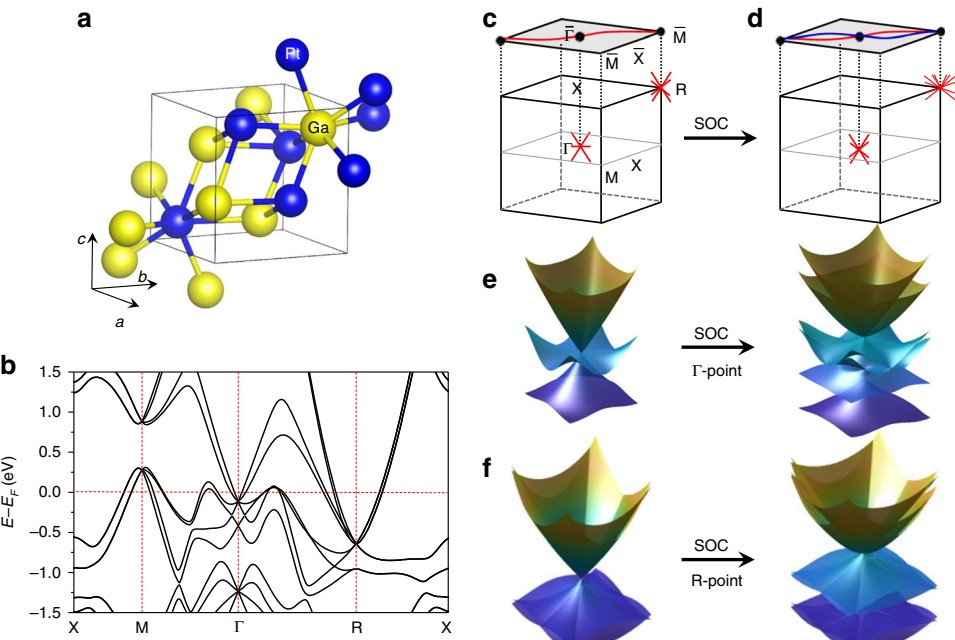

**Fig. 1 Effect of strong SOC on chiral topological states. a** Chiral crystal structure of PtGa with space group $P2_13$ (No. 198). **b** Ab initio calculated bulk-band structure of PtGa along high-symmetry lines with SOC. **c** Bulk BZ with (001) surface of PtGa along with corresponding high-symmetry points. The effect of strong SOC and spin splitting of the topological states in the whole BZ of PtGa (**d**), and the multifold fermions at the Γ (**e**) and R point (**f**).

(001) Fermi-surface (FS) projection demonstrate the existence of hole and electron pockets at $\bar{\Gamma}$ and $\bar{M}$, respectively, resulting in Fermi arcs to span over the entire BZ surface. Subsequently, giant spin splitting of the said Fermi arcs was observed post SOC incorporation, as depicted in Fig. 1d. Consequently splitting also occurs for the bulk bands. Owing to the crystal symmetry, a threefold degenerated point, the location of which coincides with that of Γ, follows the spin-1 Hamiltonian and hosts a topological charge of Chern number +2. Likewise, band degeneracy at the location of momentum R yields a double-Weyl fermion with Chern number −2, thereby causing the entire system to follow the "no-go theorem" (Fig. 1c, e, f). Considering SOC, the spin-1 fermionic excitation at Γ splits into a doubly-degenerated Weyl and fourfold degenerated Rarita–Schwinger–Weyl fermions with topological charges of Chern number +4. Meanwhile, the double-Weyl fermions at R transform into sixfold fermionic points with Chern numbers −4 as well as a trivial double-degenerated point (Fig. 1b, d–f). Therefore, it can be inferred that PtGa serves as an ideal platform to visualize the effect of strong SOC on quasi-particle excitations of multifold fermions in chiral-topology semimetals.

**Fermi-surface topology from quantum oscillation.** Quantum oscillations in single-crystalline PtGa were investigated in this study to detect spin-split FS pockets due to strong SOC, as illustrated in Fig. 1b, e, and f. Magnetic field up to 7 T was used to perform isothermal magnetization measurements at various temperatures. Results obtained for the case $B||[001]$ are depicted in Fig. 2a. Clear de Haas–van Alphen (dHvA) quantum oscillations were observed starting, at 1.8 K, from a field of ~0.5 T, thereby indicating the high quality of the grown single crystal and low effective mass of the associated charge carriers. In addition, a low Dingle temperature of ~2.64 ± 0.19 K was observed (Supplementary Fig. 7). Interestingly, characteristics of quantum oscillations observed for cases $B||[001]$, $B||[110]$, and $B||[111]$ were radically different, thereby reflecting the high anisotropy of the PtGa Fermi surface (Supplementary Figs. 4, 5). A smooth background was subtracted from the measured magnetization data and oscillations periodic in $1/B$ are clearly resolved up to 15 K

(Fig. 2b and Supplementary Fig. 6). We analyzed the corresponding fast Fourier transformation (FFT) to determine the dHvA frequencies and, for simplicity, only the 2-K data are depicted in Fig. 2d. Corresponding dHvA oscillations and FFT results obtained for all intermediate temperatures are shown in Supplementary Fig. 6.

The different dHvA frequencies in the FFT results were ascribed to extremal areas of the FS by constructing the full FS with the help of ab initio calculations, details of that are discussed in the Methods section. Evidently, the energy bands of PtGa get spin-split owing to the strong SOC and non-centrosymmetric crystal structure and this results in the emergence of FS pairs having similar shapes but largely different sizes. The complete FS along with the BZ is illustrated in Fig. 2e, revealing that the FSs are mainly spread around the high-symmetry locations Γ, R, and M. In this study, all dHvA frequencies obtained via FFT could be easily matched with calculated $k$-space extremal FS cross-section areas with their normal vectors along the $z$-axis. For simplicity, however, FSs corresponding to only the Γ-point are depicted in the inset of Fig. 2d. The identified FSs for the other FFT frequencies are presented in Supplementary Fig. 8. The fourfold band crossing at the Γ-point generates two electron types near-spherical FSs (with extremal frequencies of 31.9 T for the $\alpha_1$ and 93.96 T for the $\alpha_2$ orbit) and two square-box-shaped FSs. For each of this latter FS, two extremal cross-sections were determined that match well with the experimental FFT results in the frequency range depicted in Fig. 2d ($\beta_1$ at 113.22 T and $\beta_2$ at 259.14 T; $\gamma_1$ at 492.6 T and $\gamma_2$ at 679.37 T). The observed large frequency difference of 187 T for the latter pair clearly reveals the giant spin splitting of FSs due to strong SOC. The effective mass $m^*$ of the various spin-split Fermi pockets is estimated from the temperature dependence of the corresponding dHvA frequencies (Fig. 2c) using the thermal damping factor of the Lifshitz–Kosevich formula, $R_T = X/\sinh(X)$. Here, $X = 14.69\, m^* T/B$ and $B$ is the magnetic field averaged over $1/B$. The extremal cross-sectional areas $A_F$, Fermi wave vector $k_F$ and Fermi velocity $v_F$ of the FS pockets shown in Fig. 2d are estimated using the Onsager relation $F = (\Phi_0/2\pi^2)A_F$, where $\Phi_0 = h/2e$ (=$2.068 \times 10^{-15}$ Wb) is the magnetic flux quantum with $h$ the Planck constant; $k_F = \sqrt{A_F/\pi}$,

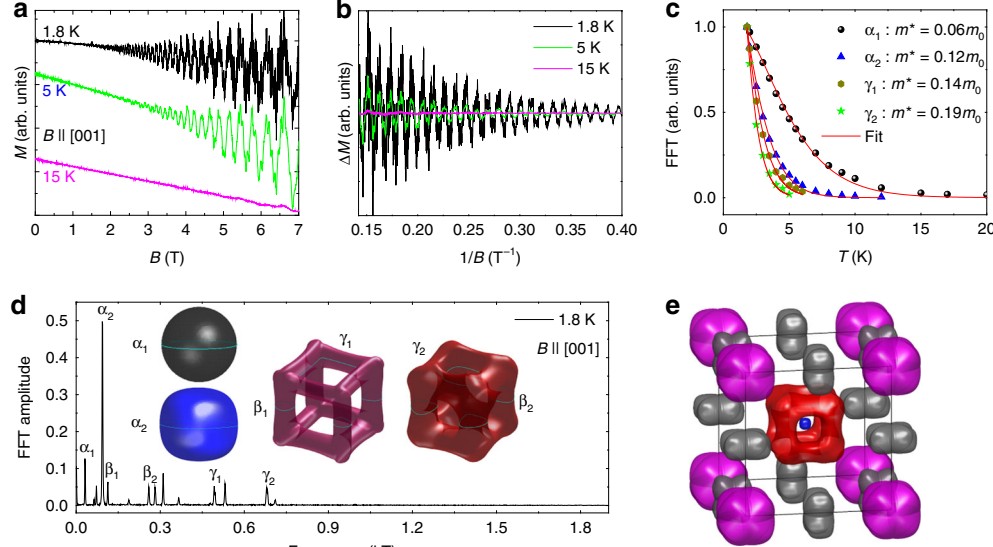

**Fig. 2 Detection of spin-split Fermi-surface pockets with quantum oscillations. a** Isothermal magnetization with dHvA quantum oscillations observed at different temperatures for $B||[001]$. **b** Corresponding dHvA signal after subtracting a smooth background. **c** Temperature dependence of FFT amplitudes for selected peaks illustrated in **d**. The Fermi-surface pockets with the identified extremal areas are shown in the inset of **d**. **e** Overall 3D Fermi-surface pockets combined in the first BZ.

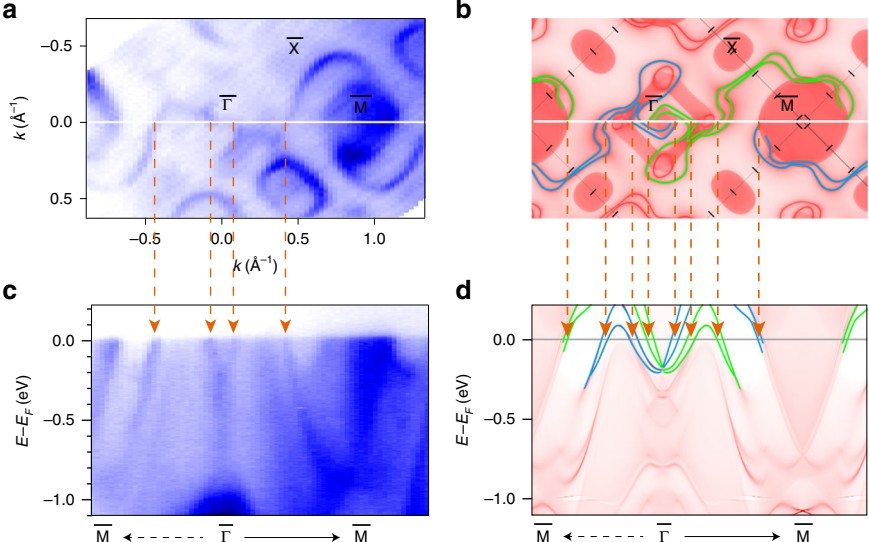

**Fig. 3 PtGa band structure. a, b** photon energy of, $h\nu = 67$ eV. Calculated Fermi arcs are highlighted with green and blue lines. **c, d** ntensity plot is acquired with $h\nu = 60$ eV. Orange arrows indicate the crossing positions of chiral Fermi arcs with $E_F$. Both the calculated constant energy contour and the band structure are rigidly shifted by 100 meV to match the ARPES data.

$\nu_F = k_F h/2\pi m^*$. The estimated $m^*$, $A_F$, $k_F$, and $\nu_F$ values of various spin-split Fermi pockets are summarized in Supplementary Table 2. Interestingly, the cyclotron masses of the FS pockets in PtGa are much lighter compared with the chiral sister compound CoSi (~1.2 $m_0$) and are similar to the well-known Weyl semimetal-like TaAs[27,28].

**Angle-resolved photoemission spectroscopy.** Using low-energy high-resolution ARPES on the high-quality single crystal, we investigate the electronic band structure of PtGa. An FS intensity plot was obtained for the 1st BZ along with ARPES intensity plots along high-symmetry directions on the (001) surface, as depicted in Fig. 3a and c. The calculated FS is presented in Fig. 3b, whereas, the band structure combining surface and bulk states are shown in Fig. 3d. By comparing the ARPES spectrum against results obtained from ab initio calculations—including the electron band at $\overline{\Gamma}$ and electronic pocket at $\overline{M}$—it is evident that the ARPES spectrum is well reproduced by calculated SSs. Owing to the relatively low photon energy, no bulk states were observed in the ARPES spectrum. As observed via our calculations, four spin-split surface bands correspond to Fermi-arc-related states that originate from the $\overline{\Gamma}$ point. Two of these bands (green lines) extend to the $\overline{M}$ point at the right side, while the other two (blue lines) connect the left $\overline{M}$ point. The experimental data are in good agreement with the calculations. In Fig. 3c, four crossing points are observed between $\overline{M}$–$\overline{\Gamma}$–$\overline{M}$, as indicated with orange arrows along the white line of Fig. 3a. Each crossing point contains two spin-split Fermi arcs. However, due to the finite ARPES resolution, the spin splitting of the Fermi arcs are difficult to distinguish along the $\overline{M}$–$\overline{\Gamma}$–$\overline{M}$ direction. Compared with the experiments, the calculated Fermi arcs have much more twisted paths, resulting more crossing points.

**Spin-split Fermi arc.** We closely look into the PtGa SSs to realize the giant SOC-induced spin splitting of the Fermi arc. On the (001) surface, the Fermi arc was observed to cross the entire BZ that helically wraps around the two $\overline{M}$ point with opposite chirality. Figure 4a depicts a 'z'-shaped Fermi arc with 60 eV photon energy, connecting the electron pocket on upper right corner and

the one on the left bottom. We present four constant energy maps in Fig. 4b to highlight the split Fermi arcs. The corresponding calculated constant energy maps at different binding energies are presented in Supplementary Fig. 9. In panel (i) of Fig. 4b, two arcs separate near $\overline{M}$ at $\Delta E = E - E_F = -0.1$ eV. With decreasing binding energy, the inner arc (orange) detaches from the outer Fermi arc (yellow) and the separation becomes larger, as shown in (ii) of Fig. 4b with $\Delta E = -0.2$ eV. Further decrease of the binding energy with $\Delta E = -0.3$ eV, the orange inner Fermi arc disappears as depicted in panel (iii). Finally, the yellow Fermi arc becomes shorter with $\Delta E = -0.4$ eV in panel (iv) of Fig. 4b. To demonstrate the splitting of the Fermi arc, we acquired two ARPES spectra along two cuts, as indicated by the white and green lines in Fig. 4a. Because of the strong SOC in PtGa, we observed a distinct splitting of the Fermi arc from the band dispersion. The largest energy difference of the splitting in Fig. 4c and d is ~0.2 eV. We also calculated the surface spin texture for $\Delta E = E - E_F = -0.1$ eV as shown in Supplementary Fig. 10. It is evident that each pair of neighboring surface states shows different spin textures, with almost opposite spin orientations. This indicates that each pair of neighboring surface states are split from one state.

In Fig. 4e and f, we show the band structures along loop 1 and 2, as indicated in Fig. 4a. The loop 1 enclosing $\overline{M}$, presents two right-moving surface band crossings $E_F$, as indicated with black arrows. The band splittings is clearly observed in Fig. 4c, d and also shown in Fig. 4e. Therefore, each crossing contains two surface bands, suggesting a chiral charge of the $\overline{M}$ point as $|C| = 4$. The $\overline{\Gamma}$ point is enclosed by loop 2, which shows six-band crossings, including four left-movings and two right-movings. Here the right- and left-moving crossings denote opposite chiral charge. Therefore, one pair of right- and left-moving crossings cancels out and does not contribute to the chiral charge. Since each crossing contains two spin-split Fermi arcs, the net crossing count along loop 2 is four; and the resulting chiral charge of the $\overline{\Gamma}$ point is $|C| = 4$. This is the first experimental observation of SOC-induced spin-split Fermi arcs and the verification of the maximal Chern number of 4 in topological chiral crystals. Since multifold fermions are a generic feature of all the chiral topological crystals with no. 198 SG, there, indeed, exist four Fermi-arc SSs connecting the $\overline{\Gamma}$ and $\overline{M}$ points.

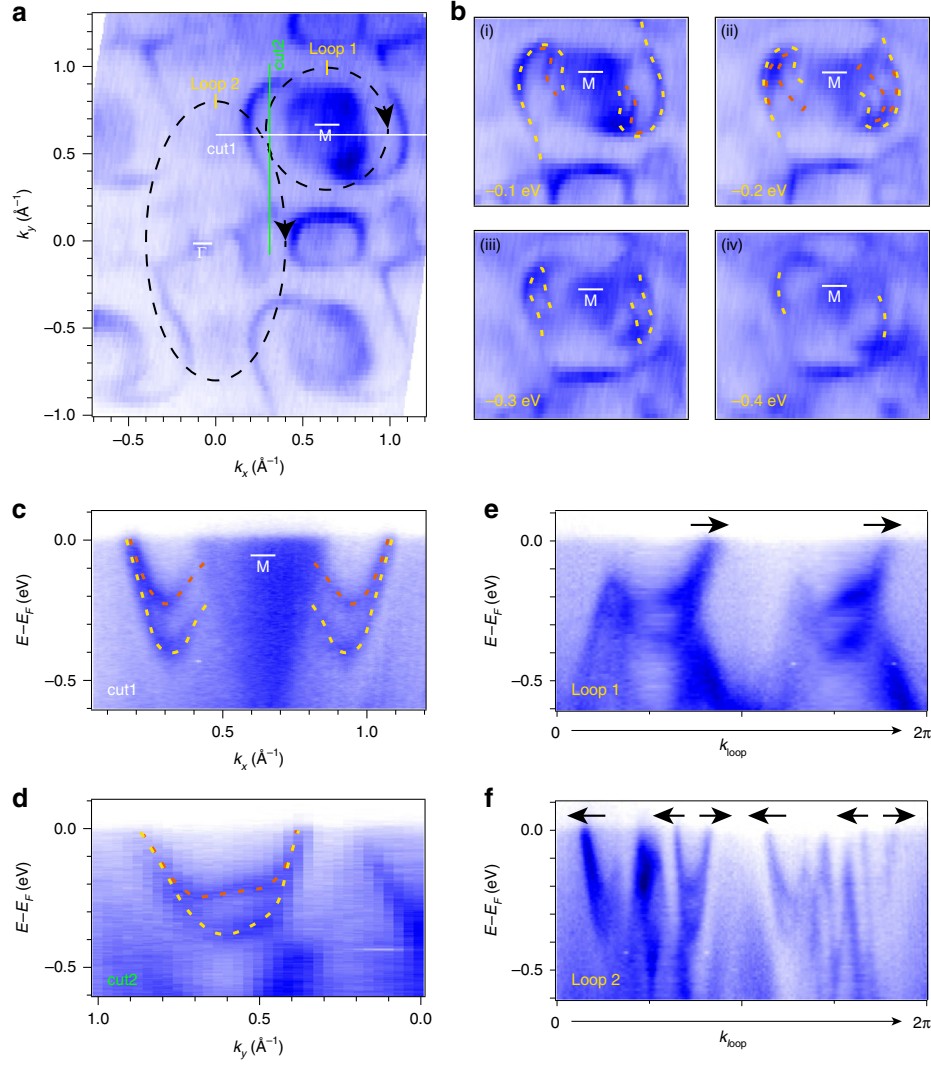

**Fig. 4 SOC-induced spin-split Fermi arcs. a** FS intensity plots obtained with photon energy, $h\nu = 60$ eV. **b** Series of constant energy maps with different binding energies, (i)-(iv) $\Delta E = E - E_F = -0.1, -0.2, -0.3$, and $-0.4$ eV, respectively. The yellow and orange dashed lines are guide-to-eye of the Fermi arcs. **c**, **d** ARPES spectra, obtained with photon energy, $h\nu = 60$ eV, along the $k_x$ and $k_y$ directions across the Fermi arcs, indicated by the white and green lines respectively in **a**. **e**, **f** ARPES spectra along loop 1 and loop 2, respectively. The paths of the loops are shown in **a**, starting from the yellow marks and proceeding clockwise. The black arrows are indications of Fermi-arcs' Fermi velocity.

## Discussion

In conclusion, this paper presents a comprehensive investigation performed using high-resolution ARPES, quantum-oscillation measurements, and state-of-the-art ab initio calculations to examine a giant SOC-induced splitting of Fermi arcs and bulk states. Owing to its large SOC, chiral PtGa demonstrates strong spin splitting, as observed via both dHvA quantum-oscillation analysis and surface ARPES measurements. The splitting of Fermi arcs connecting the time-reversal invariant points of $\overline{\Gamma}$ and $\overline{M}$ directly confirms a Chern number of $\pm 4$ for chiral multifold fermions that exist in this class of topological materials. Thus, the proposed study confirms realization of a maximal Chern number equal to $\pm 4$ for the first time in multifold fermionic systems. SOC can be considered as an effective parameter for tuning the sharpness of surface Fermi arcs, thereby paving the way for further observation and exploration of Fermi-arc-related phenomena in multifold chiral fermions.

## Methods

**PtGa crystal growth and structural refinement**. PtGa single crystals were grown from the melt using the self-flux technique. A polycrystalline ingot was first prepared by arc melting stoichiometric amounts of constituent metals with 99.99% purity in an argon atmosphere. Subsequently, the single-phase ingot was crushed, placed in an alumina crucible, and sealed within a quartz tube. The assembly was then melted inside a commercial box-type furnace at 1150 °C and maintained at that temperature for 10 h to ensure homogeneous mixing of the melt. The sample was then slowly cooled to 1050 °C at a rate of 1 °C/h followed by further cooling to 850 °C at a rate of 50 °C/h. Finally, the sample was annealed at 850 °C for 120 h prior to being cooled to 500 °C at a rate of 5 °C/h. As observed, the annealing process has a major impact on the quality of the grown crystal. In electrical transport, the RRR value significantly increases (up to ~80) with post annealing compared with the corresponding value (~24) for fast-cooled single crystals. The obtained single crystal measured ~6 mm in diameter and 17 mm in length, as depicted in Supplementary Fig. 1. Single crystallinity of the grown crystal was first verified at room temperature using a white-beam backscattering Laue X-ray setup. Observation of of single single, sharp Laue spots clearly revealed the excellent quality of the grown crystal sans any twinning or domains. A representative Laue pattern, superposed with a theoretically simulated one is also depicted in Supplementary Fig. 1. Chemical composition of the PtGa crystal was verified via energy-dispersive X-ray (EDX) spectroscopy, results of which demonstrated good agreement with the target composition of PtGa. To analyze structural chirality of the grown crystal, rigorous single-crystal X-ray diffraction experiments were performed, results of which are discussed in Supplementary Information Note 1 structural characterization section. The refined crystal structure confirmed Form A in ref. [29] only, as indicated by a Flack parameter value of ~0.03(4), which is indicative of a single-handed domain.

**Magnetic and electrical transport measurements**. The magnetization measurements were performed using a commercial MPMS3 from Quantum Design. In electrical transport, the longitudinal and Hall resistivity were measured using a low-frequency ACT option in a physical property measurement system (PPMS-9T, Quantum Design).

**Angle-resolved photoemission spectroscopy**. ARPES experiments were carried out at the Berliner Elektronenspeicherring für Synchrotronstrahlung (BESSY) (beamline UE112-PGM-1) with a Scienta Omicron R8000 analyzer, and at beamline 13U of the National Synchrotron Radiation Laboratory (NSRL) with a Scienta Omicron DA30 analyzer. The single-crystalline samples were oriented and finely polished on the (001) surfaces. The samples were Ar-ion sputtered for 30 min in ultra-high vacuum chamber, and then annealed at 680 °C for 30 min.

**First-principles calculations**. The electronic structure calculations were performed based on density functional theory (DFT) by using the full-potential local-orbital code[30] with a localized atomic basis and full-potential treatment. The exchange and correlation energies were considered in the generalized gradient approximation (GGA) level[31]. We have projected the Bloch wave functions into the atomic-orbital-like Wannier functions, and constructed the tight binding model Hamiltonian. With the tight binding model Hamiltonian, the SSs were calculated in a half-infinite boundary condition using the Green's function method[32,33].

## Data availability

The supporting data that is used to illustrate the findings of this study are available on request from the corresponding authors K.M. and C.F. upon request.

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

## Acknowledgements

The work in MPI-CPFS Dresden was financially supported by the European Research Council (ERC) Advanced Grant No. 291472 "Idea Heusler" and 742068 "TOP-MAT"; European Union's Horizon 2020 research and innovation program (grant No. 824123 and 766566) and Deutsche Forschungsgemeinschaft (Project-ID 258499086 and FE 633/30-1). S.N.G. thanks the Alexander von Humboldt Foundation for fellowship. B.S., J.H., S.C., and J.W. acknowledge the support from the DFG through the Würzburg-Dresden Cluster of Excellence on Complexity and Topology in Quantum Matter -ct.qmat (EXC 2147, project-id 39085490), the ANR-DFG grant 'Fermi-NESt', and by HLD at HZDR, member of the European Magnetic Field Laboratory (EMFL).

## Author contributions

K.M. conceived the idea. All the authors in the paper contributed to the intellectual content of this project. The work was guided by M.Y., K.M., Y.S., and C.F. PtGa single crystal was grown by K.M. and analyzed the crystal structure, chirality etc. with the help of H.B. The ARPES experiments were carried out and analyzed by M.Y. with the help of A.F., V.V., Z.S., and J.F. All the theoretical calculations and analysis were performed by Q.Y. and Y.S. K.M. carried out the quantum-oscillation experiment and analyzed the data with the help of B.V.S., J.H., S.C., and J.W. The electrical transport expt. was carried out by K.M. The paper was written by M.Y., K.M., and Y.S. with the help of C.S., N.K., S.N.G., and C.F.

## Competing interests

The authors declare no conflict of interest.
