## [Peer Review File · Nature Communications]

Reviewers' comments:

Reviewer #1 (Remarks to the Author):

The authors of the manuscript report quantum oscillations, ARPES and DFT results on PtGa, a chiral topological system. They performed quantum oscillations measurements to obtain the Fermi surfaces, and found agreement between the experimentally measured and theoretically calculated dispersions. They also showed surface states measured by ARPES. Based on these observations, they claim that they have evidence for strong SOC chiral topological system with Chern number = 4 .

Recently chiral topological nodes chiral system is a hot topic in the field of topological materials. Experimental verification of fermion with topological charge 4 would be important in the field. If what they state in the manuscript are correct, I would recommend publishing the manuscript. However, I cannot be convinced in current status. My comments is listed.

1. The major problem with this manuscript is lack of evidence of chirality of surface states. In the ARPES spectra in fig 4 d, two bands pass through the fermi level twice and one passes through once. Therefore, only the right one can be identified as chiral surface state. But to prove that the charge is equal to 4, two bands, which pass through E_f odd times, should be observed. To show the chirality of the surface states better, it would be better if spectra along a closed-loop cut can displayed, like in ref. 4 and 6.
2. In fig 4 b, the guiding line (both yellow and red) of fermi arc forms a closed loop on the left side of M. But only trivial states can have the loop. The chiral fermi arc must be fully open.
3. Good dHvA quantum oscillations data. In FFT, more than six peaks are shown. The others should be characterized.
4. (minor point) The quality of fig. 3(b) is too low to show k_z -independent. Also, it is not important if authors can prove the chirality of the observed bands. It is not a necessary map.

Reviewer #2 (Remarks to the Author):

The manuscript by Yao et al. presented evidence for "giant spin-split Fermi-arc with maximal Chern number in chiral topological semimetal PtGa" using angle-resolved photoemission spectroscopy, quantum-oscillation measurements, and DFT calculations. Topological chiral crystals including both Fermi arcs, Chern number, and bulk multifold fermions have been reported already in a number of compound. So the new thing here is the "giant" spin splitting. I have reservations both about the novelty and the data quality of this paper. Before those are addressed, I cannot recommend this paper for publication

About novelty, I agree with the authors that the "giant" spin splitting has not been observed in previous cases. However, the authors failed to explain why observing the spin splitting would be important or significant.

Now about data. I have the following concerns.

1. Generally, many data have been masked by the guides to the eyes. For instance, in Fig. 3a, the authors drew the green lines, but it is very hard for me to judge if there are indeed clear bands beneath those green lines.
2. The authors argue for giant spin splitting, but there is no spin-resolved ARPES data.
3. Somehow the spin splitting is not visible in the surface Fermi surface (Fig. 4a) at all. As shown by the white dotted lines in Fig. 4a, there are only 2 Fermi arcs originating from the Γ point. Now, the number of Fermi arcs from Γ should be 2 (if we ignore spin splitting) but 4 (if we consider spin splitting). So the fact that the authors are seeing only two Fermi arcs from Γ means that they do not observe any spin splitting from the surface Fermi surface in Fig. 4a.
4. Connected to point #3, Fig. 3c and Fig. 4a seem to be inconsistent. In Fig 3c, from Γ to M , we can at least count 8 Fermi crossings. But in Fig. 4a, if I draw a straight line from Γ to M , and count the number of Fermi crossings, there seems to be much fewer. In general, the ARPES data were not presented in a systematic way. It is very hard for me to make sense of both the band dispersions in Fig. 3 and the constant energy contours in Fig. 4 in a consistent way to reconstruct the band structure.
5. There is no DFT calculation of the surface Fermi surface and constant energy contours to compare with the data in Fig. 4.

Reply to the reviewers' comments for the manuscript "Observation of giant spin-split Fermi-arc with maximal Chern number in chiral topological semimetal PtGa"

Reviewer #1

Reviewers' comment Q1: The authors of the manuscript report quantum oscillations, ARPES and DFT results on PtGa, a chiral topological system. They performed quantum oscillations measurements to obtain the Fermi surfaces, and found agreement between the experimentally measured and theoretically calculated dispersions. They also showed surface states measured by ARPES. Based on these observations, they claim that they have evidence for strong SOC chiral topological system with Chern number = 4.

Recently chiral topological nodes chiral system is a hot topic in the field of topological materials. Experimental verification of fermion with topological charge 4 would be important in the field. If what they state in the manuscript are correct, I would recommend publishing the manuscript.

Authors reply: We thank the reviewer for taking the time to read our manuscript and to provide thorough comments, which have helped us to improve the manuscript. The reviewer accurately summarizes the key points of our experiment. We are glad that the reviewer recognizes the novelty and significance of our work.

Reviewers' comment Q2: However, I cannot be convinced in current status. My comments is listed.

1. The major problem with this manuscript is lack of evidence of charity of surface states. In the ARPES spectra in fig 4 d, two bands pass through the fermi level twice and one passes through once. Therefore, only the right one can be identified as chiral surface state. But to prove that the charge is equal to 4, two bands, which pass through E_f odd times, should be observed. To show the charity of the surface states better, it would be better if spectra along a closed-loop cut can displayed, like in ref. 4 and 6.

Authors reply: We thank the reviewer for this remark, which helped us to improve our statements. Following the referee’s suggestion, we have added two new closed-loop cuts around the Γ and M points in Fig. 4 a,e,f in the revised manuscript. Similar to ref. 4 and 6, we count two crossings along the loop around \bar{M} , and two net crossings around $\bar{\Gamma}$. However, different from the references, our data clearly shows each crossing including two bands. Therefore, we prove that the chiral charge of $\bar{\Gamma}$ and \bar{M} is ± 4 .

Manuscript revisions:

Following the referee’s suggestion we have updated Fig.4 in the revised manuscript. We have included the following discussion regarding the chirality of the surface states:

“In Figs. 4(e) and 4(f), we show the band structures along loop 1 and 2, as indicated in 4(a). The loop 1 enclosing \bar{M} , presents two right-moving surface band crossings E_F , as indicated with black arrows. The band splittings is clearly observed in Figs. 4(c) and 4(d), and also shown in Fig. 4(e). Therefore, each crossing contains two surface bands, suggesting a chiral charge of the \bar{M} point as $|C|=4$. The $\bar{\Gamma}$ point is enclosed by loop 2, which shows six band crossings, including four left-movings and two right-movings. Here the right- and left-moving crossings denote opposite chiral charge. Therefore, one pair of right- and left-moving crossings cancels out and does not contribute to the chiral charge. Since each crossing contains two spin-split Fermi arcs, the net crossing count along loop 2 is four; and the resulting chiral charge of the $\bar{\Gamma}$ point is $|C|=4$. This is the first experimental observation of SOC induced spin-split Fermi arcs and the verification of the maximal Chern number of 4 in topological chiral crystals. Since multifold fermions are a generic feature of all the chiral topological crystals with no. 198 SG, there, indeed, exist 4 Fermi-arc Ss connecting the $\bar{\Gamma}$ and \bar{M} points.”

Reviewers' comment Q3: 2. In fig 4 b, the guiding line (both yellow and red) of fermi arc forms a closed loop on the left side of M. But only trivial states can have the loop. The chiral fermi arc must be fully open.

Authors reply: We again thank the reviewer for pointing out this mistake and our wrong interpretation of the experimental data. We had an in-depth look at the calculated constant energy maps to understand the dispersion of the Fermi arcs. Indeed the chiral arcs are open as we describe in the revised manuscript. We revised the guiding lines in Fig. 4b. A comparison of theoretical and experimental data is shown below revealing the open Fermi-arc with clear spin-splitting. We have also added the calculated constant energy contours as Fig. S-9 in the supplementary materials. The surface states obtained from ARPES and theoretical calculations do not exactly overlap due to the difference in details of the surface potential considered in the theoretical calculations. However, they show the same topology and similar shape, since the surface state is decided by the topological bulk band order.

Reviewers' comment Q4: 3. Good dHvA quantum oscillations data. In FFT, more than six peaks are shown. The others should be characterized.

Authors reply: Indeed we characterized all the peaks observed in the dHvA oscillations with the help of *ab-initio* calculations. Following the referee's suggestion

we have included a new figure as Fig. S-8 in the supplementary information with identified Fermi surfaces for the additional FFT peaks. Since the main aim of the manuscript is to focus on the giant SOC-induced splitting of surface states, we focused our discussion on the most important spin-split Fermi-surface pockets at the Γ -point.

Manuscript revisions:

Following the referee's suggestion we have included the following figure in the SI.

Fig. S-8: Additional spin-split Fermi pockets with identified extremal areas at different points of the Brillouin zone for different bands as shown in Fig. S-4. The corresponding frequencies are indicated in the FFT spectrum of the dHvA oscillations in the upper panel.

Reviewers' comment Q5: 4. (minor point) The quality of fig. 3(b) is too low to show kz-independent. Also, it is not important if authors can prove the charity of the observed bands. It is not a necessary map.

Authors reply: We thank the reviewer for this advice. We intended to prove the surface origin of the Fermi-arcs with Fig. 3b. We have removed it from the revised manuscript.

Reviewer #2

Reviewers' comment Q1: The manuscript by Yao et al. presented evidence for “giant spin-split Fermi-arc with maximal Chern number in chiral topological semimetal PtGa” using angle-resolved photoemission spectroscopy, quantum-oscillation measurements, and DFT calculations. Topological chiral crystals including both Fermi arcs, Chern number, and bulk multifold fermions have been reported already in a number of compound. So the new thing here is the “giant” spin splitting. I have reservations both about the novelty and the data quality of this paper. Before those are addressed, I cannot recommend this paper for publication.

Authors reply: We thank the reviewer for taking the time to read our manuscript and provide important comments, which have helped us to improve the manuscript. Indeed, as the referee correctly illustrated, the formation of multifold fermions with large Fermi arc etc., have already been reported in several recent publications. The main novelty of the present work is the first time observation of Fermi-arc splitting and the verification of the maximal Chern number 4 in multifold fermionic systems.

Reviewers' comment Q2: About novelty, I agree with the authors that the “giant” spin splitting has not been observed in previous cases. However, the authors failed to explain why observing the spin splitting would be important or significant.

Authors reply: We thank the reviewer for appreciating the significance of our work. Here we would like to take the opportunity to explain why our findings of a spin-split Fermi surface constitute a breakthrough and open new lines of research.

The multifold fermions, identified in previous works (such as PRL 122, 76402 (2019); Nature 567, 500–505 (2019); Nature 567, 496–499 (2019); Nat. Phys. 15, 759–765(2019)), act as sources and sinks of quantized Berry flux through any surface enclosing the nodes, which is proportional to their Chern number. Though it is difficult to measure the bulk topological invariant directly, it is possible to infer its magnitude from the Fermi-arcs that are connecting these multifold fermions with opposite Chern number. For space group 198, the multifold fermions are located at

the center and corners of the Brillouin zone (Γ and R points, respectively). Due to protected time-reversal symmetry and the bulk-boundary correspondence, the Chern number is equivalent to the number of edge states (or the two-dimensional quantum Hall phase), associated with any imaginary two-dimensional slice dividing the Brillouin zone between the Γ and R points. By considering an ensemble of 2D slices that fill the entire 3D Brillouin zone, one can see that the number of edge states corresponds to the number of Fermi-arcs. Hence, by counting the number of Fermi-arcs one can estimate the Chern number of the associated fermions.

Significance of the Chern number 4: In the last decade, there have been several investigations on the possible maximal Chern number in various quadratic and cubic band crossings, like PRL 108, 266802 (2012), PRB 96, 45102 (2017) *etc.* It is found that a Chern number up to $|C| = 3$ is possible in such cases. However, recent investigations (Science 353, 5037 (2016); PRL 119, 206401 (2017); PRL 119, 206402 (2017); Nat. Mater. 17, 978–985 (2018) *etc.*) predicted that chiral multifold compounds with 198 SG can show the maximal Chern number 4. However, recent experimental studies on such chiral topological semimetals (PRL 122, 76402 (2019); Nature 567, 500–505 (2019); Nature 567, 496–499 (2019); Nat. Phys. 15, 759–765(2019)) failed to resolve the spin-split Fermi-arcs due to the small spin-orbit-coupling (SOC) and assigned the fermions with $|C| = 2$. Here, for the first time, we observe clear spin-split Fermi-arcs in PtGa which contains the strongest SOC among all known topological chiral crystals. Our investigation is the first experimental report for the maximal Chern number $|C| = 4$ nature can provide. This is why our finding is such a breakthrough in the field of topological materials, because many of the exotic phenomena predicted for topological semimetals, such as the strength of quantized photocurrents, the number of chiral Landau levels, or the number of edge modes at domain boundaries, are directly proportional to the Chern number.

We would like to point out that our results are universal for other multifold chiral fermion compounds with space group 198, since their effective Hamiltonians are

constrained by the same symmetries as PtGa. This universality provides a strong motivation to work further on the materials science aspect of this field - e.g. by doping or alloying different chiral materials, or by synthesizing new materials in space group 198 – to engineer the band structure for investigations with other probes, such as optical as-well-as transport experiments *etc.*

Reviewers' comment Q3: Now about data. I have the following concerns.

1. Generally, many data have been masked by the guides to the eyes. For instance, in Fig. 3a, the authors drew the green lines, but it is very hard for me to judge if there are indeed clear bands beneath those green lines.

Authors reply: We thank the reviewer for the advice to improve the readability of our manuscript. We have removed the concerned guide lines and rearranged Fig. 3 with experimental and calculated constant energy contours for comparison to clearly indicate the crossing positions of the chiral Fermi arcs.

Manuscript revisions:

We have modified Fig. 3 and the associated discussion in the revised manuscript so as to clearly visualize the Fermi arc dispersions. Now the portion reads as:

Using low-energy high resolution ARPES on the high-quality single crystal, we investigate the electronic band structure of PtGa. An FS intensity plot was obtained for the 1st BZ along with ARPES intensity plots along high-symmetry directions on the (001) surface, as depicted in Figs. 3(a) and 3(c). The calculated FS is presented in Fig. 3(b), whereas, the band structure combining surface and bulk states are shown in Fig. 3(d). By comparing the ARPES spectrum against results obtained from *ab initio* calculations—including the electron band at $\bar{\Gamma}$ and electronic pocket at \bar{M} —it is evident that the ARPES spectrum is well reproduced by calculated SSs. Owing to the relatively low photon energy, no bulk states were observed in the ARPES spectrum. As observed via our calculations, four spin-split surface bands correspond to Fermi-arc-related states that originate from the $\bar{\Gamma}$ point. Two of these bands (green

lines) extend to the \bar{M} point at the right side, while the other two (blue lines) connect the left \bar{M} point. The experimental data are in good agreement with the calculations. In Fig. 3(c), four crossing points are observed between $\bar{M}-\bar{\Gamma}-\bar{M}$, as indicated with orange arrows along the white line of Fig. 3(a). Each crossing point contains two spin-split Fermi arcs. However, due to the finite ARPES resolution, the spin-splitting of the Fermi-arcs are difficult to distinguish along the $\bar{M}-\bar{\Gamma}-\bar{M}$ direction. Compared to the experiments, the calculated Fermi-arcs have much more twisted paths, resulting more crossing points.

Fig. 3: **PtGa band structure.** (a),(b) Experimental and calculated constant energy contour plots at the Fermi energy with high symmetry points labelled. The ARPES data are obtained with a photon energy of $h\nu = 67$ eV. Calculated Fermi arcs are highlighted with green and blue lines. (c),(d) Experimental and calculated band structure along the $\bar{M}-\bar{\Gamma}-\bar{M}$ direction, which are indicated with white lines in (a) and (b), respectively. The ARPES intensity plot is acquired with $h\nu = 60$ eV. Orange arrows indicate the

crossing positions of chiral Fermi arcs with E_F . Both the calculated constant energy contour and the band structure are rigidly shifted by 100 meV to match the ARPES data.

Reviewers' comment Q3: The authors argue for giant spin splitting, but there is no spin-resolved ARPES data.

Authors reply: PtGa has the largest spin orbit coupling among all the CoSi family of compounds investigated so far. Strong spin orbit coupling splits one Fermi arc into two spin-split Fermi-arcs, changing the chiral charge of the connecting Fermi-arc accordingly. In our work, for the first time we have successfully observed the spin-split Fermi-arcs with a comprehensive study of high-resolution ARPES and state-of-the-art *ab initio* calculations, giving direct evidence of the chiral charges of the $\bar{\Gamma}$ and \bar{M} points. Beyond that, a spin-resolved ARPES experiment is a highly demanding experiment involving a long waiting time due to the very limited beamline-access and, therefore, is beyond the scope of the present report. We thank the reviewer for understanding our limitation. However, we have performed a calculation for the surface state spin texture. As shown below, one can see that each pair of neighboring surface states shows different spin textures with almost opposite spin orientations. This indicates that each pair of neighboring surface states are derived from one state without considering SOC. We have included the figure (Fig.

S-10) and the analysis in the revised manuscript.

Fig. S-10: Calculated spin texture of the surface state at $\Delta E = E - E_F = -0.1$ eV.

Reviewers' comment Q4: Somehow the spin splitting is not visible in the surface Fermi surface (Fig. 4a) at all. As shown by the white dotted lines in Fig. 4a, there are only 2 Fermi arcs originating from the $\bar{\Gamma}$ point. Now, the number of Fermi arcs from $\bar{\Gamma}$ should be 2 (if we ignore spin splitting) but 4 (if we consider spin splitting). So the fact that the authors are seeing only two Fermi arcs from $\bar{\Gamma}$ means that they do not observe any spin splitting from the surface Fermi surface in Fig. 4a.

Authors reply: We thank the reviewer for highlighting this crucial point. We have modified Fig. 4 to improve our manuscript allowing for a better understanding and clarification. There are in total four Fermi-arcs originating from the $\bar{\Gamma}$ point and extending to two \bar{M} points. However, the band splitting is difficult to be observed at E_F . That's why it looks as if only two bands originate from $\bar{\Gamma}$, instead of four bands. In order to resolve this issue, we present Fig. 4(b-d) to show the clear band splitting.

Furthermore, we have added Fig. 4(e,f) to show the band dispersions along two loops enclosing \bar{M} and $\bar{\Gamma}$. In Fig. 4(e), we see that the bands cross E_F twice, and each crossing contains two Fermi arcs. Therefore, the two crossings suggest that the chiral charge of the \bar{M} point is $|C| = 4$. In Fig. 4(f), we observe that the bands cross E_F six times, including four left-moving and two right moving. One pair of left- and right-moving cancel out. Therefore, the net crossings in loop 2 around the $\bar{\Gamma}$ point is two. Since each crossing contains two Fermi arcs, the resulting chiral charge of $\bar{\Gamma}$ is $|C| = 4$.

Reviewers' comment Q5: 4. Connected to point #3, Fig. 3c and Fig. 4a seem to be inconsistent. In Fig 3c, from Gamma_bar to M_bar, we can at least count 8 Fermi crossings. But in Fig. 4a, if I draw a straight line from Gamma_bar to M_bar, and count the number of Fermi crossings, there seems to be much fewer. In general, the ARPES data were not presented in a systematic way. It is very hard for me to make sense of both the band dispersions in Fig. 3 and the constant energy contours in Fig. 4 in a consistent way to reconstruct the band structure.

Authors reply: We thank the reviewer for this valuable advice. We have revised Figs. 3 and 4 and the associated text to present a consistent and coherent analysis of the ARPES data to improve our manuscript. The experimental data and calculations are presented side-by-side for comparison in the revised Fig. 3. Because the band splitting is hard to distinguish at E_F along this direction, we observed only four band crossings along the $\bar{M}-\bar{\Gamma}-\bar{M}$ direction. On the other hand, the calculated Fermi arcs have a slightly different and more twisted path than the ARPES data due to details of the surface potential difference. Therefore, more crossings can be seen along the $\bar{M}-\bar{\Gamma}-\bar{M}$ direction in the calculated pattern, as shown in Fig. 3(d). Nevertheless, the different Fermi-arc path does not change the nature of topology.

Reviewers' comment Q6: 5. There is no DFT calculation of the surface Fermi surface and constant energy contours to compare with the data in Fig. 4.

Authors reply: The DFT calculation of the surface Fermi surface and constant energy contours are shown below, with the same scale as used in Fig. 4a. We have included DFT calculated constant energy contour plots at the Fermi energy in Fig. 3 and constant energy maps with different binding energies in Fig. S-9 of the revised manuscript. As we have explained, the surface states obtained from ARPES and theoretical calculations do not exactly overlap due to the difference in details of the surface potential considered in the theoretical calculations. However, they show the

same topology and similar shape, since the surface state is decided by the topological bulk band order.

REVIEWERS' COMMENTS:

Reviewer #1 (Remarks to the Author):

I have read the revised manuscript. The authors improved it a lot and clearly demonstrated that $C=4$ in PtGe. I recommend it for publication.

Reviewer #2 (Remarks to the Author):

I appreciate the strong efforts that the authors made in order to address the questions. I would recommend the revised paper for publication now.